# On the Thermodynamics of Particles Obeying Monotone Statistics

**DOI:** 10.3390/e25020216

**Published:** 2023-01-22

**Authors:** Fabio Ciolli, Francesco Fidaleo, Chiara Marullo

**Affiliations:** 1Dipartimento di Matematica e Informatica, Università della Calabria, Via Pietro Bucci, Cubo 30B, I-87036 Rende, Italy; 2Dipartimento di Matematica, Università di Roma Tor Vergata, Via della Ricerca Scientifica, 1, I-00133 Roma, Italy; 3Dipartimento di Matematica “Guido Castelnuovo”, Sapienza Università di Roma, Piazzale Aldo Moro 5, I-00185 Roma, Italy

**Keywords:** monotone grand-canonical ensemble, thermodynamics of grand-canonical ensemble, exclusion principle, high- and low-density regimes

## Abstract

The aim of the present paper is to provide a preliminary investigation of the thermodynamics of particles obeying monotone statistics. To render the potential physical applications realistic, we propose a modified scheme called *block-monotone*, based on a partial order arising from the natural one on the spectrum of a positive Hamiltonian with compact resolvent. The block-monotone scheme is never comparable with the weak monotone one and is reduced to the usual monotone scheme whenever all the eigenvalues of the involved Hamiltonian are non-degenerate. Through a detailed analysis of a model based on the quantum harmonic oscillator, we can see that: (a) the computation of the grand-partition function does not require the Gibbs correction factor n! (connected with the indistinguishability of particles) in the various terms of its expansion with respect to the activity; and (b) the decimation of terms contributing to the grand-partition function leads to a kind of “exclusion principle” analogous to the Pauli exclusion principle enjoined by Fermi particles, which is more relevant in the high-density regime and becomes negligible in the low-density regime, as expected.

## 1. Introduction

In recent years, the investigation of exotic models has significantly increased in the hope of making some progress in solving long-standing unsolved problems involved in the physics of complex models. In this regard, we certainly must mention the question of providing a satisfactory mathematical description of the quantum electrodynamics, which obtains predictions via the renormalisation technique that are in surprisingly perfect accordance with the experiments.

Along the same line, we can identify the models which aim to study and unify the strong interactions (i.e., quantum chromodynamics) with the electroweak ones. All these models are called “standard” models and have the same strengths and weaknesses. That is, they are in good accordance with the experiments but not satisfactory from the mathematical point of view. The long-standing problem of unifying these three fundamental forces, which are present in nature, with the remaining one, that is, the gravitation, which was recently addressed through the use of the so-called noncommutative geometry (e.g., [1]), is very far from being solved, even in a partial form.

We should also mention the potential relevance of the investigation of models enjoying exotic commutation relations with some other disciplines, such as information theory and quantum computing, both of which are connected with, and relevant for, concrete applications.

Among such models, we can certainly find those associated with the so called *q*-particles, or quons, q∈(−1,0)⋃(0,1), from the perspective of the extension to the so-called anyons, corresponding to the case when the parameter *q* assumes values in some roots of the unity, and plektons.

Such exotic *q*-particles are naturally associated with the following commutation relations: (1)$$a_{q}\left(f\right)a_{q}^{†}\left(g\right)-qa_{q}^{†}\left(g\right)a\left(f\right)=\langle g,f\rangle I_{H}\;,\;f,g\;\in \;H,$$H being the one-particle space, where the creators and annihilators act on the corresponding Fock spaces. The quons can certainly be seen as an interpolation between the particles obeying the Fermi statistics (i.e., q=−1) and those obeying the Bose statistics (i.e., q=1), passing through the q=0 value describing the classical particles and, thus, obeying the Boltzmann statistics. We can observe that for q=±1, the commutation rules (Equation 1) are reduced to the well-known ones associated with the Bose and Fermi particles, respectively (see, e.g., [2]).

All such models are relevant to the so-called *quantum probability*. In fact, the Boltzmann case q=0, describing the statistics of classical particles concerning the physical meaning, is also known as *free* because it naturally arises from a particular case of quantum probability called *free probability* (see, e.g., [3]).

In the setting of quantum probability, the various generalisations of the commutation rules (Equation 1) allow one to introduce and investigate exotic quantum stochastic processes (see, e.g., [4]).

As is well-known in the Bose and Fermi cases, all the above-mentioned commutation rules naturally arise from the so called second quantisation, which is associated with the so-called *grand-canonical ensemble*. The various functors of the second quantisation allow one to construct the corresponding Fock spaces. The Fock space encodes the statistics that the involved particles obey and allows the involved commutation relations to be faithfully represented.

We can also remark that the grand-canonical ensemble allows one to investigate one of the most fascinating phenomena occurring in the condensate state of the matter, involving bosons, that is, the *Bose–Einstein Condensation* in the fundamental state (see, e.g., [2,5]). In [6], it is shown that such a phenomenon of condensation also appears in the case of Bose-like quons, that is, when q∈(0,1].

Returning to exotic commutation relations and their applications in quantum probability, we can cite the Boolean and Monotone ones. As in the case of all the models mentioned above, they satisfy commutation relations falling into the general form described in [7] (Corollary 3.2), and are therefore associated with a suitable Fock space. The Boolean Fock space is the simplest non-trivial example of second quantisation, because only one particle can be created and/or annihilated. In fact, it describes the absorption of a single photon, at most, from an apparatus (see [8]).

The monotone statistics of particles, independently introduced in [9,10], do not seem to have any evident physical application. The arising Fock space is easily constructed, as described in Section 2, using the monotonic prescription induced by a totally ordered orthonormal basis of the one-particle Hilbert space.

Now, we can also point out the well-known deep connection between the second quantisation scheme and the equilibrium statistical mechanics (see, e.g., [2,5]). Indeed, starting from a Fock space constructed by taking into account the statistics of the involved particles, we can compute, at least in principle, the so-called grand-partition function. Such a crucial function is supposed to encode all the thermodynamic properties enjoyed by a large number (of the order of the Avogadro number $N_{A}∼10^{23}$) of involved particles. This is certainly true for the Bose and Fermi cases and also, due to its simplicity, for the boolean one, in which the indistinguishability of the particles plays no role.

Due to the Gibbs paradox (cf. [5]), the computation of the grand-partition function in the Boltzmann case, in terms of the associated full Fock space, deserves a suitable correction due to the supposed indistinguishability of the involved particles (see below). The general case of the quons (i.e., q∈(−1,0)∪(0,1)) is differently solved in [11], since the necessarily “deformed” statistics that such exotic particles obey are completely unknown.

The case of non-interacting particles obeying monotone statistics, simply called *monotone particles* in the following pages, is unclear for two reasons. The first one is that the concrete physical applications of such a model are completely unknown. The second one is that we do not know whether the monotone scheme directly encodes the principle of the indistinguishability of particles, the latter being a fundamental prescription for the development of equilibrium statistical mechanics.

Taking into account all the previous considerations, it becomes natural to address the investigation of the thermodynamic properties of particles obeying the monotone prescription, which are encoded in the monotone Fock space. Unfortunately, since there is no natural, total order of the one-particle subspace on an orthonormal basis, this investigation deserves an appropriate preliminary analysis.

A simple one-particle physical system confined in a finite volume is essentially described by a Hamiltonian *H*, which is a self-adjoint positive operator with compact resolvent acting on a separable Hilbert space H. The statistics (i.e., Bose/Fermi or Boltzmann) of very large systems formed of a number of the order of the Avogadro number of non-interacting particles is encoded in the corresponding Fock space.

Since there is a natural order of the eigenvectors of *H* induced by the corresponding eigenvalues (i.e., the energy levels of the system under consideration), one is tempted to use such an order to implement the monotone scheme for models of statistical mechanics. This can be carried out only when the eigenvalues of *H* all have a multiplicity of 1 or, in simple terms, when any energy level of the model is non-degenerate.

Unfortunately, this is not the case for all concrete models when the degeneracy of all the energy levels increases to infinity in the thermodynamic limit (i.e., when, in particular, the volume of the system tends to occupy the whole environment), in which case the so-called “passage to the continuum” can be performed (see, e.g., [5,12]).

The passage to the continuum is the fundamental tool used to investigate the thermodynamic properties of more realistic models in which the one-particle Hamiltonian has a continuum spectrum, such as a free particle living in $R^{3}$.

In the present paper, we propose a method that can be used to overcome this basic difficulty and, thus, take into account the possible degeneracy of the energy levels. Indeed, we simply generalise the monotone prescription to index sets, which are merely partially ordered according to those arising from the spectrum of a positive, compact resolvent Hamiltonian with possible degenerate energy levels. This model, called *block-monotone* in the following pages, which is expected to be more suitable for physical applications, is described in Section 3.

Since the grand-partition function of a system associated with (block-)monotone particles is not directly computable in most infinite dimensional cases, the remaining part of the paper is devoted to a detailed analysis of a simple model formed of infinite uncoupled quantum harmonic oscillators. Since the corresponding Hamiltonian has non-degenerate eigenvalues, such a computation falls into the monotone scheme. The grand-partition function for such a model is computed in Section 4.

Section 5 is then devoted to the explicit computation of the statistical weights appearing in the expansion of the monotone grand-canonical partition function and to a refined study of the high- and low-density regimes. Such an investigation leads to the following relevant facts.

First of all, the computation of such statistical weights suggests that the Gibbs correction factor n!, connected with the indistinguishability of particles in the various terms of its expansion with respect to the activity, directly appears in the low-density regime, that is, when the temperature of the system becomes increasingly higher. This suggests that the monotone scheme directly encodes the indistinguishability of the involved particles.

Secondly, the decimation of the terms contributing to the grand-partition function provides a kind of “exclusion principle” analogous to the Pauli exclusion principle observed for Fermi particles. Such an exclusion principle appears to be more relevant to the high-density regime and becomes negligible in the low-density regime, as expected.

The last part of Section 5 is devoted to a refined comparison between the grand-partition functions relative to the Boltzmann and monotone models, allowing us to estimate the correction of the relevant thermodynamic quantities, such as the average number, in the low-density regime.

To conclude the present introduction, we point out that this preliminary investigation seems to provide a promising perspective concerning the potential physical applications of the block-monotone scheme, which we plan to return to in a future work.

## 2. Preliminaries

**One-particle Hamiltonian.** We start with a system whose Hamiltonian *H* is a self-adjoint positive (i.e., σ(H)⊂[0,+∞)) operator with compact resolvent, acting on a separable Hilbert space H, called the *one-particle space*.

In such a situation, the spectrum σ(H) is formed of isolated points, accumulating at +∞ if H is infinite dimensional. In addition, the multiplicity g(ε) of each eigenvalue ε∈σ(H) is finite. In summary, by considering the resolution of the identity of *H*, we obtain $I\equiv I_{H}=\sum_{\varepsilon \in \sigma (H)}P_{\varepsilon }$,
$$H=\sum_{\varepsilon \in \sigma (H)}\varepsilon P_{\varepsilon },\;\;\mathrm{and}\;\;g\left(\varepsilon \right)=\dim\left(\mathrm{Ran}(P_{\varepsilon })\right)<\infty \;.$$

Let $k_{B}\approx 1.3806488\times 10^{-23}\;JK^{-1}$ be the Boltzmann constant, and $\beta :=\frac{1}{k_{B}T}$ the “inverse temperature”. Assuming that $e^{-\beta H}$ is trace class for each β>0, we can define the *partition function* $\zeta :=\mathrm{Tr}(e^{-\beta H})$.

**The grand-partition function.** Here, we define the the grand-partition function in a relatively general framework relative to a gas comprising non-interacting particles obeying rather general statistics and thus potentially suitable for physical applications. The knowledge of such grand-partition functions plays a crucial role in the so-called *equilibrium statistical mechanics*. The standard method for such an analysis is the so-called *second quantisation*, (see, e.g., [2,5]).

Indeed, for the one-particle Hilbert space H, we define the so-called *full Fock space*
$F_{0}\left(H\right)\equiv F$, given by
$$F:=\overset{+\infty }{\underset{n=0}{⨁}}\underset{n-\mathrm{times}}{\underbrace{H⊗\cdots ⊗H}}\;,$$
with the convention that $\underset{0-\mathrm{times}}{\underbrace{H⊗\cdots ⊗H}}:=C\equiv C\Omega$, where Ω is the so-called *vacuum vector*. The *number operator N* has a clear meaning (see e.g., [2]).

For a linear operator *A* with domain D⊂H, we define
$$\begin{array}{cc} d\Gamma_{o}{\left(A\right)\lceil }_{D⊗\cdots ⊗D}:= & A⊗I⊗\cdots ⊗I+I⊗A⊗\cdots ⊗I\\ +\cdots + & I⊗\cdots ⊗A⊗I+I⊗\cdots ⊗I⊗A\;, \end{array}$$
and extend it to the whole Fock space by linearity. If *A* is self-adjoint, the closure dΓ(A) of $d\Gamma_{o}\left(A\right)$ will still be self-adjoint (see, e.g., [2]). Note that $d\Gamma (I_{H})$ provides the number operator.

Now, let *P* be a self-adjoint projection acting on F. For a fixed positive operator *H* (i.e., a Hamiltonian) and the parameters β>0 (the inverse temperature) and μ∈R (the chemical potential), such that $Pe^{-\beta d\Gamma (H-\mu I)}P\equiv Pe^{-\beta (d\Gamma (H)-\mu N)}P$ is trace class, we define the *grand-partition function* as
(2)$$Z_{P}\equiv Z_{H,P}\left(\beta ,\mu \right):=\mathrm{Tr}\left(Pe^{-\beta d\Gamma (H-\mu I)}P\right)\;.$$

The most important cases, describing the thermodynamics of Bose and Fermi gases, are those when *P* is the self-adjoint projections onto the completely symmetric and antisymmetric subspaces (with respect to the natural action of the permutations on F), respectively.

When *P* is the identity operator $I\equiv I_{F}$, the corresponding grand-partition function will describe the thermodynamics of classical particles, that is, those obeying the Boltzmann statistics. Unfortunately, this is not the case, as explained in [11,13]. Below, we outline how it is possible to recover such a grand-partition function in the Boltzmann case.

We can also remark that the *q*-deformed Fock space $F_{q}\left(H\right)$ (e.g., [4,14] for the arising ergodic properties) could be used to compute the grand-partition function for the so-called *quons* and, thus, their thermodynamics. Unfortunately, in this case, too, the second quantisation method does not work. The grand-partition function for the free gas of quons is entirely computed in [11] without using the *q*-deformed Fock space.

The so-called *Boolean* (e.g., [15]) and *monotone* (see below) models might also be described as outlined above. This is certainly true for the Boolean case, in which $F_{\mathrm{boole}}\left(H\right)=C⨁H$ and, thus, there is no question about the indistinguishability of the particles. We will see, in the forthcoming analysis, that this is also the case for the monotone model and its generalisations addressed in the present paper.

**Monotone Fock space.** For the reader’s convenience, we outline some basic facts regarding monotone Fock spaces and their fundamental operators (see [9,10,16] for more details). Here, we define a generalisation of the monotone particles which is suitable for physical applications. However, as a particular case arising directly from quantum physics, we study, in some detail, the thermodynamics of a simple model satisfying the monotone statistics/commutation relations. For the interested reader, we point out the existence of new investigations concerning exotic commutation relations that are related to those previously mentioned and might have potential physical applications (see [17]).

For k≥1, denoted by $I_{k}:=\left\{\left(i_{1},i_{2},\ldots ,i_{k}\right)|i_{1}<i_{2}<\cdots <i_{k},\;i_{j}\in N,\;j=1,\cdots ,k\right\}$, the class of all the ordered sequences of natural numbers are of length *k*. For k=0, we take $I_{0}:=\left\{\emptyset \right\}$. If k≥0, the Hilbert space $H_{k}:=l^{2}\left(I_{k}\right)$ is called the *k*-particles space. In particular, the 0-particle space $H_{0}=l^{2}\left(\emptyset \right)$ is identified with the complex scalar field C. The monotone Fock space is then defined as $F_{m}:={⨁}_{k=0}^{\infty }H_{k}$.

With any increasing sequence $\alpha =\{i_{1},i_{2},\ldots ,i_{k}\}$ of natural numbers, we canonically associate the vector $e_{\alpha }$, which, for all such sequences α, provides the canonical basis of $F_{m}$. For each pair of such sequences $\alpha =\{i_{1},i_{2},\ldots ,i_{k}\}$, $\beta =\{j_{1},j_{2},\ldots ,j_{l}\}$, we state that α<β if $i_{k}<j_{1}$. By convention, $I_{0}<\alpha$ for each $\alpha \neq I_{0}$.

In other words, if $\left\{e_{n}|n\in N\right\}$ is the canonically ordered basis of $\ell^{2}\left(N\right)$ (or any ordered basis of a separable Hilbert space), the *n*-particle space is generated by the vectors $e_{\alpha }\equiv e_{j_{1}}⊗e_{j_{2}}⊗\cdots ⊗e_{j_{n}}$ whenever $\alpha =\{j_{1},j_{2},\cdots ,j_{n}\}$ with $j_{1}<j_{2}<\cdots <j_{n}$. If we relax the last condition by merely assuming that $j_{1}\leq j_{2}\leq \cdots \leq j_{n}$, we will obtain the so-called *weakly monotone* Fock space (see, e.g., [16]). Note that $F_{m}$ and $F_{\mathrm{wm}}$ are the range of the self-adjoint projections $P_{m}$ and $P_{\mathrm{wm}}$ acting on the full Fock space F:$$F_{m}=P_{m}F\;\;\;\mathrm{and}\;\;\;F_{\mathrm{wm}}=P_{\mathrm{wm}}F\;.$$

Even if this is not used in the forthcoming analysis, we can report the structure of the monotone creation and annihilation operators and the generating commutation relations, which are nevertheless useful for application to quantum probability. Indeed, the monotone creation and annihilation operators are, respectively, for any i∈Z, given by
(3)$$a_{i}^{†}\left(e_{\left(i_{1},i_{2},\ldots ,i_{k}\right)}\right):=\left\{\begin{array}{cc} e_{\left(i,i_{1},i_{2},\ldots ,i_{k}\right)} & \mathrm{if}\;\left\{i\right\}<\left\{i_{1},i_{2},\ldots ,i_{k}\right\}\;,\\ 0 & \; \end{array}\right.$$
otherwise,
(4)$$a_{i}\left(e_{\left(i_{1},i_{2},\ldots ,i_{k}\right)}\right):=\left\{\begin{array}{cc} e_{\left(i_{2},\ldots ,i_{k}\right)} & \mathrm{if}\;k\geq 1\;\;\;\;\;\;\mathrm{and}\;\;\;\;\;\;i=i_{1}\;.\\ 0 & \; \end{array}\right.$$

One can verify that $\left|\right|a_{i}^{†}\left|\right|=\left|\right|a_{i}\left|\right|=1$ (see e.g., [16], Proposition 8). Moreover, $a_{i}^{†}$ and $a_{i}$ are mutually adjoint and satisfy the following relations:$$\begin{array}{cc} a_{i}^{†}a_{j}^{†}=a_{j}a_{i}=0 & \mathrm{if}\;\;i\geq j\;,\\ a_{i}a_{j}^{†}=0 & \mathrm{if}\;\;i\neq j\;. \end{array}$$

In addition, the following commutation relation, designated in the weak operator topology of $B(F_{m})$ and falling into the general class of commutation relations managed in Corollary 3.2 of [7] for applications in quantum probability,
$$a_{i}a_{i}^{†}=1\;I-\sum_{k\leq i}a_{k}^{†}a_{k}$$
is also satisfied.

For some properties of monotone systems, including their ergodic properties, see, e.g., [18] and the literature cited therein.

**The grand-partition function for the Boltzmann case.** The Boltzmann (or classical) case is very particular because, in Boltzmann statistics, the Gibbs paradox (e.g., [5]) takes place and, consequently, we should suitably correct the statistical weights.

As for the computation of $Z_{\pm 1}$ in the Bose and Fermi cases (e.g., [2,5]), it might be natural to use the full Fock space F(H) and the *grand-canonical Hamiltonian*K:=dΓ(H)−μN, as explained above, provided that *K* is trace class for a fixed β and μ. This corresponds to taking $P=I_{F}$ in (Equation 2).

This easy computation is reported in [13], obtaining
$$\mathrm{Tr}\left(e^{-\beta K}\right)=\frac{1}{1-\zeta e^{\beta \mu }}\;,$$
which still holds for μ<minσ(H).

For the reasons explained in [11,13], such a formula is unrealistic. However, the correct formula should be $Z_{0}=e^{\zeta e^{\beta \mu }}$.

After defining the *fugacity*, also denoted as the *activity*, through $z:=e^{\beta \mu }$, we have:(5)$$\mathrm{Tr}\left(e^{-\beta K}\right)=\sum_{n=0}^{+\infty }\zeta^{n}z^{n}\;,\;0\leq z<1\;.$$

It is interesting to see that, if one corrects (Equation 5) with the weight n! in the denominator of the series, thus taking into account the indistinguishability of particles, as it is customary to avoid the Gibbs paradox, we obtain the correct formula:(6)$$Z_{0}=\sum_{n=0}^{+\infty }\frac{\zeta^{n}}{n!}z^{n}=e^{\zeta z}\;.$$

**Harmonic oscillator.** Since we provide a detailed study of the simple model formed of non-interacting harmonic oscillators, for the reader’s convenience, we report some basic facts that are used in the following analysis. Consequently, relative to the Boltzmann statistics, we compute the relative grand-partition function at the inverse temperature β and activity $z=e^{\beta \mu }$, μ being the chemical potential.

Indeed, given that *K* acts as the Hook strength of the spring and *m* as the mass of the involved particle, it is well-known that the spectrum of the Hamiltonian of this model consists of non-degenerate eigenvalues, given by:(7)$$\sigma \left(H\right)=\left\{\hbar \omega (n+1/2)|n\in N\right\}\;,$$
where $\omega :=\sqrt{K/m}$ is the given frequency.

In this way, the partition function is given by:$$\zeta \equiv \mathrm{Tr}\left(e^{-\beta H}\right)=e^{-\frac{\beta \hbar \omega }{2}}\sum_{n=0}^{+\infty }{\left(e^{-\beta \hbar \omega }\right)}^{n}=\frac{e^{-\frac{\beta \hbar \omega }{2}}}{1-e^{-\beta \hbar \omega }}=\frac{e^{\frac{\beta \hbar \omega }{2}}}{e^{\beta \hbar \omega }-1}=\frac{1}{2\sinh\left(\frac{\beta \hbar \omega }{2}\right)}\;.$$

After taking into account the Gibbs correction (e.g., [5,11,13]), for the grand-partition function, we obtain:$$Z_{0}=e^{z\zeta }=\sum_{n=0}^{+\infty }\frac{z^{n}e^{\frac{n\beta \hbar \omega }{2}}}{n!}{\left(\frac{1}{e^{\beta \hbar \omega }-1}\right)}^{n}\;.$$

## 3. Block-Monotone Particles

The present section is devoted to a generalisation of the monotone prescription for the statistics of the particles which, on the one hand, is more suitable for potential physical applications and, on the other hand, is always different from the weak monotone scheme briefly outlined above.

For such a purpose, we consider an index set *I*, being necessarily finite or countable, which is a finite or countable disjoint union of finite sets. Indeed, $I:={⨆}_{j=0}^{+\infty }I_{j}$, where $|I_{j}|<+\infty$, j=0,1,⋯. The set *I* is naturally partially ordered, because if $k_{j},l_{j}$ are in the same subset $I_{j}$, there is no pre-assigned order between them. Conversely, if $k_{1}\in I_{j_{1}}$ and $k_{2}\in I_{j_{2}}$, then $k_{1}≺k_{2}\Leftrightarrow j_{1}<j_{2}$.

Such a picture is suggested by the potential physical applications. In fact, a positive Hamiltonian *H* with compact resolvent acting on a separable Hilbert space H, as described in Section 2, induces a natural order, as shown above, on the natural basis of H, formed of the eigenvectors associated with the eigenvalues $\left\{\varepsilon_{j}\right\}$ of *H*, where the finite cardinality of the involved subsets are given by the degeneracies $g_{j}$ of the eigenvalues $\varepsilon_{j}$. The picture arising from this analysis is defined as *block-monotone.* The corresponding block-monotone Fock space $F_{\mathrm{bm}}$ and the relative creation and annihilator operators are easily constructed as follows below.

Let $\left\{e_{j}|j\in I\right\}$ be an orthonormal basis of H equipped with the previously described partial order. Typically, such a partial order is induced by a positive Hamiltonian with compact resolvent. As noted above, $F_{\mathrm{bm}}$ is a subspace of the full Fock one $F\equiv F_{0}$, and on the *n*-particle subspace, the *n*-particle block monotone subspace is generated as follows below.

Such an *n*-particle subspace is generated by all the sequences of the elementary (orthonormal) tensors $e_{k_{1}}⊗\cdots e_{k_{n}}$ with the condition $k_{1}<k_{2}<\cdots <k_{n}$ relative to the partial order defined above. The block monotone creator and annihilator operators assume the same form as in (Equation 3) and (Equation 4), respectively, according to the above partial order.

We denote $P_{\mathrm{bm}}$ as the self-adjoint projection acting on the full Fock space projected onto $F_{\mathrm{bm}}$. This allows us to compute the grand-partition $Z_{\mathrm{bm}}$ according to (Equation 2). We can now explicitly compute such a grand-partition for the simplest non-trivial finite dimensional case, where H is generated by the orthonormal basis $\left\{\left(e_{1},e_{2}\right),e\right\}$, the eigenvalues of a Hamiltonian whose eigenvalues are h,k, with a multiplicity of 2 and 1, respectively. We express such a grand-partition function in terms of the activity $z=e^{\beta \mu }$.

In this simple situation, the block-monotone Fock space $F_{\mathrm{bm}}$ ends with the two-particle subspace and is given by:$$F_{\mathrm{bm}}=C⨁((Ce_{1}⊕Ce_{2})⊕Ce)⨁(C\left(e_{1}⊗e\right)⊕C\left(e_{2}⊗e\right))\;.$$

Correspondingly, the grand-partition function is given by:$$Z_{\mathrm{bm}}\left(z,\beta \right)=1+z\left(2e^{-\beta h}+e^{-\beta k}\right)+2z^{2}e^{-\beta (h+k)}\;.$$

We end the present section by noting that, if all the $I_{j}$ are singletons (or empty sets), the block-monotone scheme will be reduced to the usual monotone one. In the previous example, the comparison between the block-monotone scheme and the monotone one does not depend on the order that we fix on the first subset of eigenvectors of *H*, leading to:$$Z_{m}\left(z,\beta \right)=1+z\left(2e^{-\beta h}+e^{-\beta k}\right)+z^{2}(e^{-2\beta h}+2e^{-\beta (h+k)})\;.$$
In the forthcoming analysis, we study, in some generality, a non-trivial situation of this kind.

Conversely, the block-monotone scheme is never comparable with the weakly monotone one. Indeed, the weakly monotone version of the last example provides five additional elements on the basis of the two-particle subspace plus non-trivial contributions involving all the *n*-particle subspaces. Indeed,
$$\begin{array}{cc} Z_{\mathrm{wm}}\left(z,\beta \right)= & 1+z\left(2e^{-\beta h}+e^{-\beta k}\right)+z^{2}(4e^{-2\beta h}+2e^{-\beta (h+k)}+e^{-2\beta k})\\ + & \sum_{n=3}^{+\infty }z^{n}a_{n}\left(h,k,\beta \right)\;. \end{array}$$

## 4. The Grand-Partition Function for Monotone Particles

Since the explicit computation of the monotone grand-partition function is not available for most infinite dimensional cases, we reduce the matter to the simple case of the one-dimensional quantum harmonic oscillator briefly described in Section 2. The spectrum of the involved Hamiltonian (Equation 7) is formed of multiplicity-one eigenvectors. Therefore, in such a case, the block-monotone model described in Section 3 is reduced to the usual monotone one.

Denoting such a monotone grand-partition function relative to the quantum harmonic oscillator as $Z_{m}$, we obtain the following:

**Proposition** **1.**
*For the grand-partition function $Z_{m}$, we have*

$$Z_{m}=1+\sum_{n=1}^{+\infty }z^{n}e^{\frac{n\beta \hbar \omega }{2}}\prod_{k=1}^{n}\frac{1}{\left(e^{k\beta \hbar \omega }-1\right)}\;.$$


*In addition, $0\leq Z_{m}\leq Z_{0}$, where $Z_{0}$ is the Boltzmann grand-partition function given in (Equation 6), and thus $Z_{m}$ converges for all z≥0 and β>0.*


**Proof.** We start the second half by noting that:
$$e^{k\beta \hbar \omega }-1=\left(e^{\beta \hbar \omega }-1\right)\sum_{l=0}^{k-1}e^{l\beta \hbar \omega }\geq k\left(e^{\beta \hbar \omega }-1\right)\;,$$
and, thus,
$$0\leq \prod_{k=1}^{n}\frac{1}{\left(e^{k\beta \hbar \omega }-1\right)}\leq \frac{1}{n!{\left(e^{\beta \hbar \omega }-1\right)}^{n}}\;.$$Therefore, $Z_{m}\leq Z_{0}<+\infty$.Concerning the first half, taking into account the exclusion rule arising from the monotone assumption, it is straightforward to verify that, for the contribution relative to the *n*-particle subspace, n≥1,
$$\begin{array}{cc} \mathrm{Tr} & \left(P_{m}e^{-\beta d\Gamma (H)}P_{m}\lceil_{\underset{n-\mathrm{times}}{\underbrace{H⊗\cdots ⊗H}}}\right)=e^{-\frac{n\beta \hbar \omega }{2}}\sum_{k_{1}=0}^{+\infty }e^{k_{1}\beta \hbar \omega }\sum_{k_{2}=k_{1}+1}^{+\infty }e^{k_{2}\beta \hbar \omega }\\ \; & \cdots \cdots \sum_{k_{n}=k_{n-1}+1}^{+\infty }e^{k_{n}\beta \hbar \omega }=e^{\frac{n\beta \hbar \omega }{2}}\prod_{k=1}^{n}\frac{1}{\left(e^{k\beta \hbar \omega }-1\right)}\;. \end{array}$$□

Now, we can compare both grand-partition functions $Z_{m}$ and $Z_{0}$. Indeed, after defining
$$Z_{\#}=\sum_{n=0}^{+\infty }a_{n}^{\left(\#\right)}\left(\beta \right)z^{n}\;,$$
# standing for “0” and “monotone”, we can address two physically interesting, cases, β↓0 (high-energy/low-density regime) and β↑+∞ (low-energy/high-density regime).

For the high energy case,
(8)$$a_{n}^{\left(m\right)}\left(\beta \right)=\frac{1}{{\left(\hbar \omega \right)}^{n}}\left(\frac{1}{\beta }\right)\left(\frac{1}{2\beta }\right)\cdots \left(\frac{1}{n\beta }\right)+o\left(\frac{1}{\beta^{n}}\right)\equiv \frac{1}{n!{\left(\hbar \omega \beta \right)}^{n}}+o\left(\frac{1}{\beta^{n}}\right)\;,$$
which leads to
(9)$$a_{n}^{\left(0\right)}\left(\beta \right)=a_{n}^{\left(m\right)}\left(\beta \right)+o\left(\frac{1}{\beta^{n}}\right),\;\;\mathrm{for}\;\;\beta ↓0\;.$$

**Remark** **1.**
*Equations (Equation 8) and (Equation 9) explain that, in some sense, at least in this simple case of the harmonic oscillator, in the limit of high energies (i.e., β↓0), the monotone grand-partition function $Z_{m}$ behaves, term-by-term, in the same way as $Z_{0}$, corresponding to classical particles. In the next section, we provide a more refined analysis concerning this fact.*

*For the appearance (i.e., (Equation 8)) of the Gibbs correction term n!, we can immediately argue that the monotone Fock space will naturally take into account the indistinguishability of particles.*


Now, we move on to the limit of low energies (typically when dealing with the so-called ground states) β↑+∞. Indeed,
$$\prod_{k=1}^{n}\frac{1}{\left(e^{k\beta \hbar \omega }-1\right)}\approx \frac{1}{e^{\beta \hbar \omega \sum_{k=1}^{n}k}}=e^{-\beta \hbar \omega \frac{n(n+1)}{2}}\;,$$
and, thus,
$$Z_{m}\approx \sum_{n=0}^{+\infty }z^{n}e^{-\frac{n^{2}\beta \hbar \omega }{2}}\;.$$

Relative to the Boltzmann partition function, according to the reasoning above, we obtain:$$Z_{0}\approx \sum_{n=0}^{+\infty }\frac{z^{n}}{n!}e^{-\frac{n\beta \hbar \omega }{2}},\;\;\mathrm{for}\;\;\beta ↑+\infty \;.$$

Now, using the Stirling formula, when n↑+∞ and β↑+∞, retaining only the leading terms, we obtain:(10)$$a_{n}^{\left(0\right)}\left(\beta \right)\approx e^{-n(\beta \hbar \omega /2+\ln n)},\;\;\mathrm{whereas}\;\;a_{n}^{\left(m\right)}\left(\beta \right)\approx e^{-n^{2}\beta \hbar \omega /2}\;.$$

**Remark** **2.**
*In the high-density regime described by β,n↑+∞, that is, when the state of the matter is very condensed, the monotone particles obey a kind of exclusion principle analogous to the Pauli exclusion principle for fermions. The heuristic Formula (Equation 10) seems to confirm the existence of such a principle.*


## 5. The Low-Density Regime

We discuss the low-density regime corresponding to β,z≈0 by showing that, in such a limit,
$$Z_{m}\left(\beta ,z\right)\approx Z_{0}\left(z,\beta \right),\;\;\mathrm{for}\;\;\beta ,z\to 0\;.$$

**Proposition** **2.**
*For the monotone and Boltzmann grand-partition functions, we have:*

(11)
$$\left(1-f\left(\beta ,z\right)\right)Z_{0}\left(z,\beta \right)\leq Z_{m}\left(\beta ,z\right)\leq Z_{0}\left(z,\beta \right)\;,$$

*where $f\left(\beta ,z\right):=\frac{z^{2}e^{\beta \hbar \omega }}{4(e^{\beta \hbar \omega }-1)}$.*


**Proof.** (sketch). With $x:=e^{\beta \hbar \omega }$ and
(12)$$\Delta_{n}\left(x\right):=1-\frac{n!}{\left(1+x\right)\left(1+x+x^{2}\right)\cdots \left(\sum_{k=0}^{n-1}x^{k}\right)}\;,\;n=1,2,\cdots \;,$$
noting that β↓0⇔x↓1, for n≥2 (for n=0,1, both coincide), we obtain:
(13)$$a_{n}^{\left(0\right)}-a_{n}^{\left(m\right)}=\frac{1}{n!{\left(x-1\right)}^{n}}\left(\Delta_{n}\left(1\right)+\Delta_{n}^{\prime }\left(1\right)\left(x-1\right)+\frac{\Delta_{n}^{\prime \prime }\left(\xi_{n}\right)}{2}{\left(x-1\right)}^{2}\right)\;.$$Here, we apply the Taylor formula with second-order Lagrange remainder; thus, $\xi_{n}$ is a certain number (obviously, depending on *n*) in the interval (1,x).Since $\Delta_{n}\left(1\right)=0$, $\Delta_{n}^{\prime }\left(1\right)=\frac{n(n-1)}{4}$ and, finally, we obtain the evidence that $\Delta_{n}^{\prime \prime }\left(\xi_{n}\right)<0$ whenever x>1 (see the Appendix A), collecting these elements together, we have:
$$a_{n}^{\left(0\right)}-a_{n}^{\left(m\right)}\leq \frac{\sqrt{x}}{n!{\left(x-1\right)}^{n}}\left(\frac{n(n-1)}{4}\left(x-1\right)\right)\equiv \frac{n(n-1)}{4}\left(x-1\right)a_{n}^{\left(0\right)}\;.$$After defining $y:=\frac{z\sqrt{x}}{x-1}$, we can sum up, obtaining:
(14)$$\begin{array}{cc} Z_{0}-Z_{m}\leq & \frac{\left(x-1\right)}{4}\sum_{n=0}^{+\infty }n\left(n-1\right)\frac{y^{n}}{n!}=\frac{\left(x-1\right)}{4}y^{2}\frac{d^{2}Z_{0}}{dy^{2}}\\ \equiv & \frac{z^{2}x}{4(x-1)}e^{y}\equiv \frac{z^{2}x}{4(x-1)}Z_{0}\;. \end{array}$$Now, (Equation 14) easily leads to:
$$Z_{0}\left(1-\frac{z^{2}e^{\beta \hbar \omega }}{4(e^{\beta \hbar \omega }-1)}\right)\leq Z_{m}\;,$$
and the proof follows, according to Proposition 1. □

In order to progress to the investigation of the low-density regime, (Equation 11) reads:(15)$$\left(1-f\left(\beta ,z\right)\right)\leq \frac{Z_{m}\left(\beta ,z\right)}{Z_{0}\left(\beta ,z\right)}\leq 1\;,$$
and (Equation 15) provides a useful condition, if and only if (1−f(β,z))>0.

It is now convenient to define $t:=z^{2}/4$ with the limitations 0≤t<1, obtaining:$$1-\frac{xt}{x-1}>0\Leftrightarrow x>\frac{1}{1-t}\;.$$
By passing to the logarithm and restoring the variables *z* and β, we obtain:$$\beta \hbar \omega >-\ln(1-{\left(z/2\right)}^{2})\;.$$

On the other hand, if for some 0<γ<1,
(16)$$\beta \hbar \omega \geq -\ln(1-{\left(z/2\right)}^{2\gamma })>-\ln\left(1-z^{2}/4\right)\;,$$
again, in terms of *x* and *t*, we have:$$x\geq \frac{1}{1-t^{\gamma }}\Leftrightarrow \frac{xt^{\gamma }}{x-1}\leq 1\Leftrightarrow \frac{xt}{x-1}\leq t^{1-\gamma }\to 0$$
whenever t↓0 or, equivalently, z↓0.

Restoring the original variables, the above computation simply means that:$$0\leq f\left(\beta ,z\right)\equiv \frac{z^{2}e^{\beta \hbar \omega }}{4(e^{\beta \hbar \omega }-1)}\leq {\left(z^{2}/4\right)}^{1-\gamma }\to 0\;,$$
when *z* and, necessarily, also β in the chosen region change to 0.

Thus, we prove the following:

**Proposition** **3.**
*For each fixed 0<γ<1 and (z,β) in the region R delimited by 0≤z<2 and by the condition*

$$\beta \hbar \omega >-\ln\left(1-{\left(z/2\right)}^{2\gamma }\right)\;,$$

*we obtain*

$$\lim_{\overset{\left(\beta ,z)\to (0,0\right)}{(\beta ,z)\in R}}\frac{Z_{m}\left(\beta ,z\right)}{Z_{0}\left(\beta ,z\right)}=1\;.$$



Concerning the low-density regime, Proposition 3 has the following meaning. Indeed, for β,z≈0 in the region defined by (Equation 16),
(17)$$Z_{m}\left(\beta ,z\right)\approx \left(1-f\left(\beta ,z\right)\right)Z_{0}\left(\beta ,z\right)\;,$$
where the minus sign is explained in Remark 2 as an analogue of the Pauli exclusion principle for monotone particles. Such a *monotone exclusion principle* tends to become very relevant in the high-density regime (β↑+∞, see Remark 2), whereas it tends to vanish in the low-density regime (Proposition 3). In the latter case, the function 1−f(β,z) provides the correction for the thermodynamic potentials relative to those of the monotone case, compared with the analogous ones relative to Boltzmann case.

Moreover, using (Equation 17), we can determine that the average number of particles is
$$\begin{array}{cc} N_{m}\left(\beta ,z\right)= & z\frac{\partial \ln Z_{m}\left(\beta ,z\right)}{\partial z}\approx z\frac{\partial \ln(1-f(\beta ,z))}{\partial z}+N_{0}\left(\beta ,z\right)\\ = & N_{0}\left(\beta ,z\right)-\frac{z^{2}e^{\beta \hbar \omega }}{2(\left(e^{\beta \hbar \omega }-1\right)-{\left(z/2\right)}^{2}e^{\beta \hbar \omega })}\;, \end{array}$$
where the second addendum in the last member is, indeed, negative because of (Equation 16).

We conclude by noting that Proposition 3 allows one to compute the asymptotics of *Connes’ spectral action* (cf. [1]), associated with the average number of monotone particles, as described in [19] for *q*-particles, but with the condition described in (Equation 16). We postpone the investigation of this aspect for a forthcoming analysis.

## 6. Conclusions

The previous analysis concerning monotone models suggests that their potential applications to physics appear to be meaningful and fruitful, even if the explicit computation of the thermodynamic quantities seems to be rather complicated. Therefore, it is natural to systematically address the investigation of the statistical properties of monotone systems, even if the involved grand-partition function is not completely computable for most infinite dimensional systems. We thus list some natural questions which could be addressed in future investigations below.

The first one concerns the models in which the involved energy levels are degenerate. A simple example to address is the case when the degeneracy of the energy levels is uniform. A concrete but more complicated example is the isotropic harmonic oscillator in a *d*-dimensional environment, d>1. Such a degeneracy is not uniform but can easily be computed, obtaining:ε=ℏω∑i=1d(ni+1/2),ni∈N,i=1,2,⋯,d,g(ε)=d+n−1n,n∈N.

The second natural question to be addressed is the investigation of the free gas formed of monotone particles. This is connected with the degeneracy and can be addressed either by considering the free particles confined in a box and then “passing to the continuum”, as in Sections 4 and 5 of [19], or by removing the harmonic potential (i.e., performing an appropriate limit as the Hook constant changes to 0). For d=1, the second approach seems to be directly applicable because, as pointed out above, the energy levels are non-degenerate, whereas the first approach already involves a non-trivial degeneracy in the case d=1.

The third but not least significant question is the systematic investigation of the statistical properties of monotone systems, with particular attention given to low- and high-density regimes. As explained above, the detailed investigation of the low-density regime also involves the explicit computation of the asymptotics for β↓0, which is connected to Connes’ spectral action (e.g., [1]) for monotone systems and, thus, to noncommutative geometry.

The study of the high-density regime, particularly at zero temperature (i.e., in the limit β↑∞), could explain the quantitative effect of the decimation induced by the monotone prescription. This is simply the effect that we previously called the “monotone exclusion principle”, that is, the analogue of the Pauli exclusion principle occurring in the case of fermions at zero temperature.

For example, we can argue that the statistical properties of the (block-) monotone systems might have reasonable applications to complex systems which are absorbing (or emitting) quanta of energy. The system that we have in mind is an atom which is capturing a photon and then passing into an excited state or even emitting, again, a photon reaching a more stable state. Another example concerns the nuclei of fissile material (such as uranium U${}_{235}^{92}$) in a nuclear plant which are capturing thermal neutrons and undergoing nuclear fission. In both cases, the relevant subject might not be the absorbed particles (bosons in the former case and fermions in the latter case) but the complex system which is absorbing (or emitting) the energy, according to the order of the eigenvalues of its Hamiltonian. These considerations might suggest that the block-monotone prescription has some role in the investigation of such complex systems from a statistical point of view.

We conclude by pointing out that such a monotone exclusion principle should not allow for the occurrence of the condensation phenomena of monotone particles in the fundamental state. It would interesting to provide a rigorous proof of this conjecture for the free gas of monotone particles in a *d*-dimensional (or, more concretely, in the euclidean 3-dimensional) space.

## Data Availability

Not applicable.

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
