# Peer review of "On the Thermodynamics of Particles Obeying Monotone Statistics"

_entropy, 2023, doi:10.3390/e25020216_

Round 1

Reviewer 1 Report

The mathematical level of the article exceeds most of the articles published in Entropy. I recommend adding Bożeko, M., Dołęga, M., Ejsmont, W., & Gal, Ś. R. (2022). Reflection length with two parameters in the asymptotic representation theory of type B/C and applications. Journal of Functional Analysis, 109797. -- because there is a new fock space that can be related to the article.

Author Response

Many thanks for the careful reading of our paper. We have added the suggested citation with some additional comments in lines 185-187.

Reviewer 2 Report

While it is not yet clear whether particles obeying monotone statistics are physically relevant, this paper provides interesting arguments regarding such particles. The paper is very nicely written.

In formula (3), the first condition should be $i<i_1$.  

Note that `non' cannot be used as a separate word, for example non-equilibrium, non-degenerate, etc. Also it's `a Hamiltonian', not `an Hamiltonian' 

Author Response

Many thanks for the careful reading of our paper. As suggested, we corrected everywhere the article before "Hamiltonian". In addition, after noticing that the reviewer is saying about the order of the involved indices coincides with our definition, we improved a bit the notations concerning this question (raised by the reviewer) from lines 188 up to the centered formula in line 196 before "Even if this....".

Reviewer 3 Report

 This paper deals with the thermo-statistical description of N-body Systems obeying exotic second quantization rules. This constitutes a speculative generalization of the existing theory. Nonetheless, it seems that these models and ideas have some echo in the literature. Therefore, this paper would arouse the interest of  people familiar with these currents of thought in  the audience of this journal. Hence, in my opinion, this paper meets the criteria for publication in Entropy.

Author Response

Many thanks for the careful reading of the manuscript

Reviewer 4 Report

The authors investigate the thermodynamics
of the particles obeying monotone statistics by introducing the so-called block-monotone, based on a partial order arising from the natural one on the spectrum of a positive Hamiltonian with compact resolvent. The mathematical formalism of the manuscript is well-constructed and the results sound interesting. The main criticism of this study is the lack of concrete application examples to illustrate the
pertinence of such a statistic.  
Once the authors address this issue, I will recommend the paper for publication.

Author Response

Many thanks for the careful reading of our paper. Even if the question raised by the reviewer is well-posed and interesting (as well explained in all our paper and mainly in the conclusions), it is unclear that all such exotic models (q-models, monotone, weakly monotone and many others present in literature and object of deep speculation for Quantum Probability) can have a realistic chance of some reasonable physical application. Due to the (block, weak)-monotone prescription, already in the model described in detail in our paper, any exhaustive computation of the grand-partition function is doomed to fail (except in the finite dimensional case for which we already provided some computations is Sec. 3).

On the other hand, most of our paper (Sects 4,5,7) is devoted to the simplest, but yet very complicated infinite dimensional case, confirming that the computations are very far to be exhaustive, and thus very far to provide results which are suitable for direct physical applications.

We'd like to point out again that one of the aim of our paper, combined with a detailed analysis of model based on the quantum harmonic oscillator, was just to find a reasonable route (the block-monotone prescription whose order is naturally induced by the spectrum of the Hamiltonian) in the direction of realistic physical applications.

However, we inserted few lines (320-330) of comment on the question raised by the referee.